# Computation Implemented by the Interaction of Chemical Reaction, Clustering, and De-Clustering of Molecules

**DOI:** 10.3390/biomimetics9070432

**Published:** 2024-07-16

**Authors:** Yukio Pegio Gunji, Andrew Adamatzky

**Affiliations:** 1Department of Intermedia Art and Science, School of Fundamental Science and Technology, Waseda University, Ohkubo 3-4-1, Shinjuku-ku, Tokyo 169-8555, Japan; 2Unconventional Computing Laboratory, University of the West of England, Bristol BS16 1QY, UK; andrew.adamatzky@uwe.ac.uk

**Keywords:** chemical reaction, clustering and de-clustering, oscillation, spike train, logic gate, implicit intelligence

## Abstract

A chemical reaction and its reaction environment are intrinsically linked, especially within the confines of narrow cellular spaces. Traditional models of chemical reactions often use differential equations with concentration as the primary variable, neglecting the density heterogeneity in the solution and the interaction between the reaction and its environment. We model the interaction between a chemical reaction and its environment within a geometrically confined space, such as inside a cell, by representing the environment through the size of molecular clusters. In the absence of fluctuations, the interplay between cluster size changes and the activation and inactivation of molecules induces oscillations. However, in unstable environments, the system reaches a fluctuating steady state. When an enzyme is introduced to this steady state, oscillations akin to action potential spike trains emerge. We examine the behavior of these spike trains and demonstrate that they can be used to implement logic gates. We discuss the oscillations and computations that arise from the interaction between a chemical reaction and its environment, exploring their potential for contributing to chemical intelligence.

## 1. Introduction

Theories and models of chemical reactions have traditionally assumed well-stirred solutions. However, real systems, particularly biochemical reactions within living tissues and cells, often deviate significantly from this assumption. This deviation is especially relevant when attempting to use biological or chemical materials for computing instead of silicon chips. The heterogeneity of such systems and the resulting dynamics can significantly impact computations. To explore these issues, we investigate the interaction between chemical reactions and the heterogeneity of molecule density caused by clustering (aggregation) and de-clustering (dispersion).

We also consider material-based and organism-based computations. Biochemical reactions occur within small cells and have various effects within these compact spaces. Distinctions between reactions in-water and on-water [1,2,3], and between single-molecule and multi-molecule behavior [4,5], exemplify how the interaction between chemical reactions and the intracellular environment can influence computation.

Biochemical substances and biological populations confined to small compartments, such as cells, exhibit various oscillatory phenomena. In the field of dynamical systems, oscillations are often interpreted as limit cycles [6,7]. However, in these models, variables represent the concentration of chemical substances, assuming well-stirred conditions and homogeneous density, which is inconsistent with real biological systems.

Several studies have considered the chemical reaction of the KaiC protein [8,9,10,11,12,13,14]. The phosphorylation and dephosphorylation of KaiC lead to circadian rhythms within a 24 h period. It was discovered that the KaiA, KaiB, and KaiC hexamer complex drives these rhythms [8,9,10]. Experimental evidence showed that a 24 h cycle occurs with only KaiABC and ATP in vitro [11,12]. Various models were proposed, suggesting that reaction constants depend on the structure of KaiC [13] or the hexamer/cluster of KaiC [14]. It became clear that KaiA and KaiB play roles in synchronization tuning [15], and that monomer shuffling in the hexamer during dephosphorylation also contributes to synchronization [16]. The temperature dependence of oscillation damping is expressed as a Hopf bifurcation from a limit cycle to an unstable spiral as a macroscopic analogy [17]. However, this analogy does not fully capture the role of parameters like temperature or the loss of KaiA [18].

Recent advances using real-time bioluminescence for the KaiABC system have shown detailed mechanisms of KaiABC-ATP interactions [19]. Protein structure analysis using particle accelerators indicated that circadian rhythms result from ATP hydrolysis, with phosphorylation cycles driven by ATP hydrolysis [20,21,22]. Although mathematical models incorporating these findings have been proposed [23,24], the dominant role of ATP hydrolysis in the KaiABC system is still debated. It is reported that the phase of the phosphorylation cycle can be shifted by changing the ATP/ADP ratio or adding oxidized quinone [25].

From the research history of circadian rhythms based on KaiC, we learn that the initial assumption was that gene regulation in vivo was necessary for the cyanobacterial circadian rhythm. The discovery of 24 h oscillations in vitro was surprising, suggesting a complete set of chemical substrates necessary for circadian rhythms. However, further research revealed additional factors such as the specific site of phosphorylation on KaiC, the role of KaiA, and unexpected phenomena like monomer shuffling and ATP hydrolysis oscillations.

These factors create the environment in which phosphorylation–dephosphorylation (computation) occurs, suggesting that biochemical reactions depend on various accompanying chemicals and conditions (environment for computation). This concept can be generalized to other biochemical reactions. While complex chemical reactions are programmable [26], they are not defined as closed systems but depend on the reaction environment. Novel models of synchronization based on the idea of dissipative structures have been proposed [27]. The detailed balance between individual oscillators involves energy dissipation, driving oscillator coupling, and synchronization. Each oscillator’s environment contributes to its generation, demonstrating the interaction between a chemical reaction and its environment mediated by fluctuations.

In light of these observations, we implement the interaction between a chemical reaction and its environment using an abstract chemical reaction. We assume that the reaction involves only the activation or inactivation of a chemical species and that the reaction constantly changes depending on the environment. Here, the environment is a cluster of various sizes formed by self-aggregating molecules open to fluctuations and other enzymes not initially part of the system. This situation is common in real cells. After evaluating the basic properties of the interaction between a chemical reaction and its environment, we implement logic gates based on this interaction. Since ballistic computation was proposed, various material-based and biomaterial-based logic gates have been implemented [28,29,30,31]. Finally, we discuss the relationship between intelligence inherent in cells and computing operated outside cells.

The research results presented in this paper align with the theoretical and experimental domains of enzymatic computation developed by Katz and colleagues [32,33]. However, the novelty and originality of this study lie in simulating an oscillating enzymatic system that produces outputs but does not match mechanisms, similar to glycolytic oscillations [34,35]. This innovative approach not only supports Katz’s foundational ideas but also extends them by demonstrating the potential for complex dynamic behavior in enzymatic systems, paving the way for new applications in biochemical computing and synthetic biology.

## 2. Chemical Reaction Dependent on Cluster Size

### 2.1. General Framework

The abstract chemical system is defined as $\langle M,℘\left(M\right),F_{c},\;F_{D},\;\varphi ,\;B,v\rangle$, where M is a set of molecules, $℘\left(M\right)$ is a power set of M, $F_{c}:℘\left(M\right)\times ℘\left(M\right)\to ℘\left(M\right)$ and $F_{D}:℘\left(M\right)\to ℘\left(M\right)\times ℘\left(M\right)$ are maps applied to elements of $℘\left(M\right)$, representing clustering and de-clustering of molecules, B={0,1} is the value of molecules, φ: B→B is a stochastic map controlling activation and inactivation of molecules, and v: M→B is verification for a molecule. The value of a molecule, 0 or 1, indicates inactive or active molecules, respectively.

A cluster of molecules is expressed as an element of $℘\left(M\right)$, i.e., a subset of M, and is represented by a letter that is distinguished from the other clusters. The number of molecules is preserved, and a set of molecules is divided into clusters where each cluster is mutually disjoint. A collection of all clusters is called a *partition*, P, which can be expressed as follows:(1)$$M=\left\{x\in C|\forall C\in P\right\}.$$

Given $M=\{m_{1},\;m_{2},\;\ldots ,\;m_{n}\}$, one partition is expressed as
(2)$$P=\{\left\{m_{1}\right\},\;\left\{m_{2},\;m_{3},\;\ldots ,\;m_{s}\right\},\;\left\{m_{s+1},\;\ldots \right\},\;\ldots ,\;\left\{\ldots ,\;m_{n}\right\}\}$$
where each cluster is distinguished by a letter such that
(3)$$C_{1}=\left\{m_{1}\right\},\;C_{2}=\left\{m_{2},\;m_{3},\;\ldots ,\;m_{s}\right\},\ldots ,\;C_{r}=\left\{\ldots ,\;m_{n}\right\}.$$

A cluster containing just one molecule is called a monomer. Clustering is defined for ${(C}_{p},\;C_{q})\in ℘\left(M\right)\times ℘\left(M\right)$ as follows:(4)$$F_{c}({(C}_{p},\;C_{q}))=C_{p}\cup \;C_{q},$$
with the probability $P_{C}\in [0.0,\;1.0]$ dependent on the rate of inactive molecules in a cluster such that
(5)$$P_{C}\propto \frac{\left|\left\{m\in C|v\left(m\right)=0\right\}\right|}{\left|C\right|}$$
where |S| represents the number of elements of a set S. This implies that $C_{p}$ and $C_{q}$ are chosen for clustering if both $\frac{\left|\left\{m\in C_{p}|v\left(m\right)=0\right\}\right|}{\left|C_{p}\right|}$ and $\frac{\left|\left\{m\in C_{q}|v\left(m\right)=0\right\}\right|}{\left|C_{q}\right|}$ are high enough to be chosen. In contrast, de-clustering is defined for $C_{p}\in ℘\left(M\right)$ as follows:(6)$$F_{D}(C_{p})=(C_{p_{1}},\;C_{p_{2}})$$
where $C_{p_{1}}\cup \;C_{p_{2}}=C_{p}$, how to divide the elements into two sets is determined stochastically, and de-clustering proceeds with the probability $P_{D}\in \left[0.0,\;1.0\right]$, thus satisfying the following expression:(7)$$P_{D}\propto \frac{\left|\left\{m\in C|v\left(m\right)=1\right\}\right|}{\left|C\right|}$$

The more clusters are de-clustered, the more molecules in a cluster are active. Activation and inactivation for a molecule are stochastically determined, dependent on the cluster size in which a molecule is contained. Thus, for m∈C⊆M, the following expression is generated:(8)$$\varphi \left(v\left(m\right)\right)=\left\{\begin{array}{c} \;0,\;\mathrm{if}\;v\left(m\right)=1,\;\mathrm{with}\;P_{IA}(C)\\ 1,\;\mathrm{if}\;v\left(m\right)=0,\;\mathrm{with}\;P_{A}(C) \end{array}\right.$$
where $P_{IA},\;P_{A}\in [0.0,\;1.0]$ is the probability satisfying the following conditions:(9)$$\left\{\begin{array}{c} P_{IA}\left(C\right)>P_{A}\left(C\right)\;\mathrm{if}\;\left|C\right|\;\mathrm{is}\;\mathrm{small}\;\#\;\#\\ P_{IA}\left(C\right)<P_{A}\left(C\right)\;\mathrm{if}\;\left|C\right|\;\mathrm{is}\;\mathrm{large} \end{array}\right.$$

Equations (4)–(7)—with (4) and (5) referred to as clustering conditions and (6) and (7) as de-clustering conditions—imply that clustering and de-clustering proceed depending on the rate of active or inactive molecules in a cluster. Equations (8) and (9), referred to as reaction conditions, imply that the chemical reactions of activation (from 0 to 1) and inactivation (from 1 to 0) depend on the cluster size. Throughout this paper, we assume that inactivation is dominant in monomers, while activation is dominant in large clusters.

Figure 1 illustrates the interaction between a chemical reaction and its environment, showing that the reaction proceeds within this environment. A molecule in monomer form predominantly changes from an active to an inactive state, while a molecule within a large cluster predominantly changes from an inactive to an active state.

The simulation of the interaction between the chemical reaction and its environment is conducted by the following procedure:An initial condition, consisting of the partition of molecules and the state of each molecule (active or inactive), is set.If two randomly chosen clusters satisfy the clustering condition, they combine into one cluster; otherwise, nothing occurs.If a randomly chosen cluster satisfies the de-clustering condition, it divides into two clusters; otherwise, nothing occurs.Each molecule is activated or inactivated if it satisfies the reaction condition.The partition is updated based on procedure instructions 3–4 above, which constitutes one time step.

This iterative process continues in a stepwise manner.

The chemical reaction at the molecular level is implemented using the stochastic Gillespie algorithm, which determines whether each molecule reacts based on the reaction constant [36,37,38]. Molecular-level chemical reactions can also be implemented using a multiset approach [39,40]. While the Gillespie algorithm is widely applied to stochastic chemical simulations, it does not account for the interaction between chemical reactions and molecule self-aggregation, i.e., the interaction between computation and its execution environment. In this regard, our model differs from previous molecular-based chemical reaction models.

We next consider the extreme case. Firstly, it is assumed that $P_{C}\sim 1.0$ if more than 60% of molecules in a cluster are inactive and $P_{D}\sim 1.0$ if more than 60% of molecules in a cluster are active. This tendency satisfies conditions (4)–(7). Secondly, it is assumed that $P_{IA}\left(C\right)\sim 1.0\;and\;P_{A}\left(C\right)\sim 0.0$ if C is a singleton set (i.e., monomer), and that $P_{IA}\left(C\right)\sim 0.0\;and\;P_{A}\left(C\right)\sim 1.0$ if C is a big cluster consisting of more than 90% of all molecules. It is also assumed that under other environments (i.e., various sizes of clusters), activation and inactivation are antagonistic and their ratio does not change. This assumption also satisfies conditions (8) and (9).

This extreme case shows a simple oscillation between clustering and de-clustering triggered by drastic changes in activation and inactivation. Figure 2 shows part of a time series oscillation. Here, the number of all molecules is 16 and is preserved. The initial condition is the largest cluster consisting of 16 molecules. Since $P_{IA}\left(C\right)\sim 0.0\;and\;P_{A}\left(C\right)\sim 1.0$ for a big cluster, once a big cluster is generated, all molecules are activated. Thus, the biggest cluster is broken up until all clusters become monomers. Once all clusters become monomers, they are inactivated since $P_{IA}\left(C\right)\sim 1.0\;and\;P_{A}\left(C\right)\sim 0.0$ for a monomer. Because $P_{C}\sim 1.0$ for inactive monomers, clustering proceeds until most of the molecules are collected into the biggest cluster. This process is repeated.

The yellow loop in Figure 2 depicts the trajectory of the target molecule being tracked. Initially, the target molecule is part of a cluster comprising six molecules. Subsequently, this cluster divides into two smaller clusters, each containing three molecules. Notably, the target molecule is found within one of these three-molecule clusters.

The oscillation between clustering and de-clustering remains stable under the absence of noise conditions, provided a complete initial condition is specified. This complete initial condition entails either all clusters being inactive monomers or one of the largest clusters comprising active molecules. Noise conditions are delineated by a noise level, $p_{noise}\in \left[0.0,\;1.0\right]$, whereby active molecules transition to inactive states and inactive molecules transition to active states with a specified probability, $p_{noise}$.

The oscillatory behavior of clustering and de-clustering is readily observed in the absence of noise across varying numbers of molecules. In Figure 3, the extreme case is depicted concerning the number of clusters and active molecules, with 200 molecules in total. The minimum point for the number of clusters is observed at 1, indicating the formation of the single largest cluster comprising all molecules. Subsequently, a rapid increase in the number of active molecules occurs, reaching 200. As de-clustering progresses, a substantial number of active molecules is maintained. Following this phase, as molecules transition to inactive states, clustering gradually recommences. Hence, the oscillation between clustering and de-clustering mirrors the oscillation between molecule activation and inactivation.

In the presence of noise levels ranging from 0.01 to 0.2, oscillations cease, and both the number of clusters and active molecules fluctuate, leading to a stable state, as illustrated in the lower section of Figure 3. This indicates that the oscillatory behavior of clustering and de-clustering becomes unstable under perturbed conditions. Consequently, we are prompted to explore another potential source of oscillation, namely, the contribution of enzymes.

### 2.2. Spike Oscillation Derived by Enzyme

Here, we introduce an enzyme to accelerate the chemical reaction. Given the abstract nature of our chemical reaction, we define two types of reactions: activation and inactivation. Consequently, we introduce two enzymes to accelerate these processes. Our specific focus lies on the enzyme that accelerates inactivation, as it can induce spike-like oscillations. The enzymatic activity responsible for accelerating inactivation is defined as follows:(10)$$\mathrm{If}\;0.5<\frac{\left|\left\{m\in M|v\left(m\right)=0\right\}\right|}{\left|M\right|}<P_{\alpha }\;\mathrm{and}\;v\left(m\right)=0,\;\mathrm{then}\;v\left(m\right)=1.$$

Given that the enzyme for inactivation solely facilitates the progression of inactivation, its enzymatic activity advances under conditions where inactivation predominates over activation. Therefore, the condition $0.5<\frac{\left|\left\{m\in M|v\left(m\right)=0\right\}\right|}{\left|M\right|}$ make sense. The probability $P_{\alpha }$ is variously set.

Figure 4 illustrates how the interaction between a chemical reaction and its environment under noisy conditions is significantly influenced by the presence of an enzyme, leading to spike oscillations. The noise condition is set to $p_{noise}$ = 0.02 and $P_{\alpha }=0.7$. For convenience, these parameter settings are used throughout this paper. Although it may seem counterintuitive that inactivation could increase the number of active molecules, the acceleration of inactivation promotes clustering, eventually forming the largest cluster. Once this largest cluster is generated and the conditions are met, all molecules abruptly switch from inactive to active states, resulting in a rapid increase in the number of active molecules. This predominance of active molecules leads to a gradual de-clustering process. However, due to the continuous perturbation by noise, the de-clustering is short-lived, leading to a decrease in active molecules and the resumption of clustering. As inactive molecules dominate, the enzyme facilitates clustering again, causing another drastic increase in active molecules. This cyclical process results in the appearance of spike-like oscillations.

Just as spikes are used to transmit signals in neurons, it is highly conceivable that spikes can be used as a tool for transmitting information and could serve as a prototype for computation and intelligence. This perspective opens the possibility of controlling spike formation and using the temporary administration of enzymes computationally. If logical computation can be performed externally to the cell, it is plausible that similar processes occur within the cell, indicating cellular intelligence. Therefore, we first investigate the basic behavior of spikes generated by enzyme administration.

The appearance of spikes depends on the antagonistic relationship between clustering and de-clustering and on the cluster size. However, perfect control over clustering and de-clustering is not possible. Spike generation is determined by the size of the clusters present at any given time, so even with enzyme presence, spikes may not always form successfully. When enzymes are introduced to the system, they affect both molecule activation and changes in cluster size. Consequently, the generation of spikes is inherently unstable and depends on the timing of enzyme administration.

Figure 5 shows spike generation resulting from enzyme administration. In the time series presented in Figure 5, an enzyme that accelerates inactivation is administered between 1000 and 2000 time steps. Shortly after administration, most molecules become inactive, leading to clustering and a decrease in the number of clusters. As the number of clusters continues to decrease, the number of active molecules oscillates violently in very short periods. Once the largest cluster is formed, all molecules become active, generating a spike. As de-clustering proceeds under highly activated conditions, inactivation also continues due to the enzyme, reaching an equilibrium point between clustering and de-clustering. This equilibrium then shifts back to clustering, generating a spike-like signal. Clustering continues until the largest cluster is formed, while the number of active molecules continues to oscillate rapidly.

The time series shown in the top part of Figure 5 demonstrates that after the occurrence of a series of spikes, the number of clusters remains low. This suggests that large clusters persist without disassembling. The time series shown in the bottom part of Figure 5 indicates that the second administration of the enzyme starts 1000 steps after the first. It is evident that clustering is maintained after the first enzyme administration, which is why the second administration generates a spike shortly after its initiation. It takes a very short time to reach the largest cluster. This raises the question of whether the reduction in the number of clusters is a general tendency.

Figure 6 shows how the spike train is influenced by enzymatic strength, defined by the probability of inactivation applied to a molecule satisfying condition (10). The percentages above the graph represent this probability, indicating the enzymatic strength. The second enzyme administration occurs 1000 steps after the termination of the first for all enzymatic strengths. While it was considered that higher enzymatic strength would lead to a greater reduction in the number of clusters, the data show no significant difference in cluster reduction between 100% and 20% enzymatic strengths. Although the 40% enzymatic strength shows a noticeable reduction, this result is merely a consequence of probability.

Subsequently, the behavior of spikes and changes in cluster size were investigated at even weaker enzyme strengths, as shown in Figure 7. When the enzyme strength drops below 10%, the effects become pronounced. As the number of clusters increases after the first enzyme administration, it becomes difficult to reach the minimal point (i.e., the largest cluster). Therefore, spikes are not generated smoothly after the first enzyme administration. At the 1% enzyme strength level, even the first administration fails to produce a spike train. It is evident that weaker enzyme strengths are insufficient to form the largest cluster and generate spike trains.

Even though enzyme administration contributes stochastically to the system, it is anticipated that cluster formation will be promoted, and this effect will persist even after the enzyme has dissipated. Essentially, the initial enzyme administration is memorized in the form of cluster formation. To assess this memory effect, we evaluated the duration for which the memory of enzyme administration persisted after the initial administration.

### 2.3. Memory Effect or after Effect of Chemical Reaction

Considering that weaker enzyme strengths are expected to influence the memory effect, enzyme strengths of 2% and 1% were selected to investigate this effect. The initial condition was set as a complete initial condition, with all active molecules contained within a large cluster. Concurrently, the system was consistently exposed to a 2% perturbation, resulting in fluctuated steady states if the enzyme was not administered. The first enzyme administration began at 1000 time steps and concluded at 2000 time steps. The second enzyme administration commenced at either 200, 300, 500, 700, 900, or 1100 steps after the termination of the first enzyme administration.

Figure 8 illustrates the aftereffect of enzyme administration, with the enzyme strength set at 2%. Even at this level of enzyme strength, it significantly contributes to spike train generation. The aftereffect of enzyme administration is particularly evident when the second enzyme is administered at 300 steps after the termination of the first enzyme administration. At this point, or at 500 steps after the first enzyme administration, the number of clusters remains low, indicating the persistence of large clusters due to the initial enzyme administration. Consequently, when the second enzyme administration inactivates molecules, it easily generates the largest cluster, leading to a sudden activation change in all molecules, expressed as a spike.

The aftereffect observed at 300 steps after the first enzyme administration may contribute to more spikes than those driven solely by the first enzyme administration. However, at 500 steps after the first administration, the number of clusters does not reach the standard steady-state level of about 100 molecules. In contrast, when the second enzyme is administered between 700 and 1100 time steps after the first, the number of clusters does not decrease sufficiently to generate the largest cluster, and the number of active molecules oscillates at lower levels. Notably, when the second enzyme is administered at 900, 1000, or 1100 time steps after the first enzyme administration, it takes some time to generate spikes, and only two spikes are produced. This implies a diminishment of the aftereffect.

Figure 9 presents another instance of the aftereffect of enzyme administration. Even with the first administration of the enzyme failing to reduce the number of clusters in the system, it appears that there is minimal aftereffect. However, due to the lingering effect of the first enzyme administration, the second administration can prove effective. At a 1% enzyme strength level, no spikes occur after the first enzyme administration. A series of continuous enzyme administrations can be effective, as the effect of the initial administration persists, resulting in a decrease in the number of clusters and sustained damping. If the enzyme is continuously administered without a 500 h gap, it is possible to administer the enzyme for the second time while maintaining the cluster formation effect incrementally. Consequently, spike train formation becomes possible only after the second enzyme administration. At 300 steps after the first administration, two spikes are distinctly observed, and even at 400 steps after the first, there is one definite spike, indicating an explicit aftereffect of enzyme administration.

In contrast, if the second enzyme administration occurs beyond 500 steps after the termination of the first, the number of clusters increases to the standard level of the fluctuated steady state, making it unlikely for spikes to be generated. Since the aftereffect of enzyme administration is a probabilistic process, we here estimate the probability of aftereffects with respect to the distribution of cluster numbers.

Figure 10 illustrates the distribution of cluster numbers following enzyme administration. The top left graph depicts the distribution under the no-enzyme condition, representing the fluctuated steady state. In this scenario, arbitrary time steps after the virtual termination of no-enzyme administration—100, 500, and 1000 steps—are examined across 1000 trials. Given the steady state condition, the distributions at these time steps are nearly identical. Each horizontal line represents one-fifth of the total clusters, with the peak of the distribution typically occurring around 120 clusters, which is the standard level of the steady state.

The remaining graphs display distributions under various enzyme strength conditions. Under 1% enzyme strength, the distributions at 1000 and 500 steps after enzyme administration overlap. Although the distribution at 100 steps after administration differs from those at 1000 and 500 steps, it lacks a prominent peak, indicating minimal impact on cluster reduction or generation. Conversely, at 2%, 5%, 10%, and 100% enzyme strengths, the distributions at 1000, 500, and 200 steps after administration differ, showing a bimodal distribution with a smaller peak at one cluster. This suggests that the aftereffect significantly contributes to the rapid generation of the largest cluster, persisting for at least 1000 steps after administration, and indicates that rapid activation changes are memorized by cluster size dynamics in the reaction environment.

Just as a large computer heating a small room affects subsequent computer use due to elevated temperatures, the chemical reaction and its environment described in this paper exhibit a similar memory effect. However, unlike the negative impact of temperature on subsequent computations, memorizing a chemical reaction positively influences subsequent reactions. This memory effect through the computation–environment interaction is likely a common occurrence.

The occurrence of spikes and memory capacity can facilitate information processing and logical computing. Here, we implement a logic gate using the interaction between chemical reactions and their environment.

## 3. Logic Gate Implemented by the Interaction of Activation and Clustering

If the logic gate of AND-gate and NOT-gate can be defined, any logical statements can be expressed in classical logic, i.e., Boolean algebra of which the value of any expression can be determined by either 0 or 1. The AND-gate is defined by the truth table such that 0 AND 0 = 0, 0 AND 1 = 1 AND 0 = 0, and 1 AND 1 = 1. The NOT-gate is defined by the truth table such that NOT (0) = 1 and NOT (1) = 0. In contrast, OR-gate is defined by 0 OR 0 = 0 and 0 OR 1 = 1 OR 0 = 1 OR 1 = 1. As Boolean logic is defined by an algebra closed with respect to binary operations, AND, OR, and unary operation, NOT, it can be straightforwardly verified that for any value of x, y, x OR y = NOT (x AND y). Thus, one can say that Boolean logic can be well defined by the definition of AND- and NOT-gates [41].

Figure 11 illustrates how to implement AND- and NOT-gates using the interaction of chemical reactions (activation and inactivation) and their environment (various sizes of clusters). The fluctuated steady state under the no-enzyme condition is regarded as the value 0, while an adequate number of spikes generated by enzyme administration is regarded as the value 1. Inputs are prepared either by no-enzyme or enzyme administration. As the effect of the enzyme is not deterministic, the input sometimes cannot be adequately transmitted to the gate. The occurrence of spikes is estimated by counting spikes generated by enzyme administration. The output value is prepared by either no-enzyme or enzyme administration, depending on the total number of spikes obtained from a pair of input values. If the total sum of two spike trains exceeds an adequate threshold value, one spike train is generated as the value 1, indicating the operation of an AND-gate. The values 0 and 1 can be prepared by the no-enzyme condition and enzyme administration, respectively, enabling the implementation of the AND-gate as shown in Figure 11.

Similarly, it is straightforward to implement the NOT-gate. The value 0 is prepared by the no-enzyme condition, while the value 1 is prepared by the enzyme administration condition. After preparing the input value, the number of spikes generated is counted. If the number of spikes exceeds a threshold value, the output is determined as the no-enzyme condition; otherwise, the output is determined as the enzyme administration condition, allowing the transition from 0 to 1 and from 1 to 0, representing the NOT-gate.

As mentioned before, activation–inactivation and clustering–de-clustering are stochastic processes destined to be unstable. If the input and output preparations are implemented with low enzyme strength, the memory effect of enzyme administration can influence the value’s appearance. The preparation of both input and output values cannot entail one-to-one correspondence between preparation and realization, representing the essential character of this unconventional computing.

Figure 12 demonstrates the implementation of the AND-gate through the interaction of activation and clustering. The response of the output to all possible inputs is depicted, with each input response represented by a pair of vertically arranged graphs. The orange curve illustrates the time series of activated molecules, while the blue curve represents the time series of cluster numbers.

As the AND operation is a binary operation, there are two input values: one input is received by cell-1 in the upper graph, and the other input is received by cell-2 in the lower graph. Input preparation is defined by administering the no-enzyme condition for 0 and administering the enzyme condition for 1, with the enzyme strength set to 100%. The input preparation is reflected in the number of activated molecules. Consequently, the first input value is determined by the number of active molecules within the time series enclosed by the black square in cell-0, while the second input value is determined by the number of active molecules within the time series enclosed by the black square on the left in cell-1. If a spike occurs wherein the number of active molecules within the time series enclosed by the black square in cell-0 or cell-1 exceeds 190 and reaches its maximum, it is counted as one spike. If the number of spikes exceeds 2, the input value is defined as 1; otherwise, it is defined as 0.

If the total number of spikes summed up from cell-0 and cell-1 exceeds four, the output preparation is determined as the enzyme administration condition; otherwise, it is determined as the no-enzyme condition. The enzyme is administered to the system 1000 steps after the termination of the input in cell-1. By examining the time series of the number of active molecules in cell-1, which reflects the output preparation, the number of spikes is counted. If the number of spikes exceeds two, the output is defined as 1; otherwise, it is defined as 0. The output value is determined in cell-1.

Finally, the implemented AND-gate reproduces the truth table for AND-gates. For input pairs (0, 0), (0, 1), and (1, 0), the output is 0, and for input pair (1, 1), the output is 1. It is evident that the NOT-gate is easily implemented in a similar manner, with input processing executed by a different procedure. The input and output values are the same as those for the AND-gate. After counting the number of spikes, if it exceeds two (i.e., input 1), the output value preparation is determined as the no-enzyme condition; otherwise, it is determined as the enzyme administration condition. Subsequently, the time series of the number of active molecules is used to compute the output value, and the output value is determined from the number of spikes. This reproduces the truth table for the NOT-gate.

As the number of active molecules is influenced by both chemical reactions and their environment, particularly by memory effects, logic gates implemented through the interaction between chemical reactions and their environment are unstable. However, such instability could contribute to robust and adaptable computing. Exploring this possibility will be discussed in our next paper.

Recently, logic gates implemented by biomaterials such as DNA and RNA [42,43] and by molecules [44,45] are being developed for practical applications instead of silicon gates. Together with the development of logic gates made of conductive and soft materials [46], these advancements will contribute to the practical development of soft robotics. Through this development, the significance of probabilistic computations, which can be interpreted as malfunctions observed in real living organisms, will be fundamentally questioned.

## 4. Discussion

Since Ilya Prigogine proposed the concept of dissipative structures [47], it has been argued that external perturbations contribute to the dynamic and stable behavior of systems. The Bénard cell, a typical example of a dissipative structure, results from the coupling of convection due to thermal gradients and the diffusion of viscous fluids. Thermal dissipation leads to convection and the fluctuation of fluid results in the distribution of cells. While external perturbations play an essential role in pattern formation, a specific structure with high (convection) and low (diffusion) velocities of information propagation is necessary to achieve the effects of perturbation. In reaction–diffusion systems, it is also argued that the combination of slow and fast diffusion effectively couples with nonlinear activation and inhibition. These specific conditions, which enhance the effects of perturbation, have not been generalized in complex systems.

Is the mechanism generating spike trains in our model too specific to generalize to other dissipative structures or systems influenced by external factors? Is the interaction between chemical reactions and their environment too specific for general dissipative structures? Self-aggregation and dispersion of molecules are ubiquitously found in biochemical phenomena. It is also common to observe that chemical constants vary depending on the cluster size, leading to chemical oscillations influenced by the interaction between cluster size and chemical reaction. Enzymes also frequently contribute to chemical reactions. Notably, enzyme administration leads to spike trains only if the system is perpetually perturbed, implying that dissipation plays a crucial role in generating spike trains. In this sense, our model might be considered a possible extension of the concept of dissipative structures.

Although spike trains in our model are generated by the interaction between chemical reactions, clustering–de-clustering, and enzyme administration under perturbation, they might lack the fundamental property to generate autonomous periodic oscillations, such as the negotiation of chemical reactions. The recently proposed mechanism of deviating from the detailed balance of reaction coupling derived from dissipation might be a promising candidate for the hidden mechanism behind the negotiation of chemical reactions.

This perspective may represent a stronger version of dissipative structures. The intrusion of an oscillator external to a specific oscillator can lead to oscillator–oscillator coupling. External involvement in the system could signify a stronger mechanism, such as temporal cohesion [48]. This can result in stable non-equilibrium states, akin to living systems [49]. The fact that these phenomena are products of various coincidences implies that their cause cannot be found within the system itself. Essentially, the system is connected to its external environment. We posit that this external connectivity is at the core of intrinsic intelligence, distinct from the extrinsic intelligence found in mechanical systems. One of the challenges to exploring the issue of intrinsic intelligence might be Markov blankets [50].

When we say that a system possesses intelligence, does this imply the existence of a mechanism or algorithm within the system that activates this intelligence? If we consider consciousness and mind as extensions of intelligence, then a system with consciousness and mind must have an internal mechanism expressing these attributes, which negates autonomy or free will. This is paradoxical. Precisely because the cause of intelligence is sought outside the system, the system can possess intelligence that is not merely mechanical. In this sense, the extension of the concept of dissipative structures can be regarded as intrinsic intelligence.

One such computational capability surpassing the limitations of conventional systems is illustrated by breaking the trade-off between computational universality and efficiency [51,52]. The trade-off in chemical reactions is originally proposed by Michael Conrad [53]. While a knife is universally applicable but less efficient for specific uses, a wine opener can only open wine but is highly efficient for this purpose. Generalists exhibit high universality and low efficiency, whereas specialists exhibit low universality and high efficiency. There is typically a trade-off between universality and efficiency. Elementary cellular automata [54,55] also demonstrate a definite trade-off between computational universality and efficiency, where universality is defined by the number of reachable configurations, and efficiency is defined by the average velocity to reachable configurations [51,52]. Normal cellular automata are deterministic systems not connected to the external environment. However, if cellular automata are updated asynchronously, they are compelled to use a random generator, thereby becoming connected to the outside. Asynchronous cellular automata and asynchronously tuned cellular automata have been reported to break the trade-off observed in elementary cellular automata updated synchronously [51,52]. The system proposed in this paper is expected to break that trade-off, serving as an explicit sign of intrinsic intelligence.

## 5. Conclusions

The interaction between computation and the computation execution environment was implemented in the form of a chemical reaction and its reaction environment, and its behavior was investigated through simulation. A chemical reaction is expressed by the activation and inactivation of an abstract chemical substance, and the reaction environment is expressed by the size of a cluster formed by the aggregation of chemical molecules. The interaction is defined by the dominance of inactivation for monomers and the dominance of activation for big clusters, leading to oscillations between clustering and de-clustering and between activation and inactivation only in the absence of fluctuations.

Under fluctuating conditions, the system does not exhibit oscillations; the numbers of both the clusters and active molecules settle into a fluctuating steady state. By administering an enzyme that accelerates inactivation, the system produces spike trains. Accelerated inactivation leads to the formation of large clusters, triggering sudden activation of all molecules, followed by clustering. This process iterates, resulting in spike trains.

While the behavior is probabilistic, the memory of spike trains can be encoded in cluster size. This memory effect decreases the number of clusters, maintaining large clusters and making subsequent enzyme administrations easily realize spike train formation. Simultaneously, due to its probabilistic nature, the number of clusters sometimes increases, making it challenging for subsequent enzyme administrations to generate spike trains.

Logic gates were implemented using chemical reactions and enzyme administration. By controlling the number of spikes, AND- and NOT-gates were achieved. These logic gates pave the way for implementing cellular automata based on chemical reactions. Given that our system relies on various phenomena and probabilities, it remains connected to the outside environment. We contend that this external connection could represent intrinsic intelligence rather than the mechanical intelligence inherent in machines. By constructing cellular automata within this system, one could assess whether it breaks the trade-off between universality and efficiency, offering a potential verification method. Our system could be one of the candidates exhibiting implicit intelligence.

Our model does not encompass all possible cellular reactions to external stimuli. The limitations of our model and the specific conditions under which it can be applied will be a topic of further studies. Right now, our model serves as an abstract representation to generate hypotheses rather than a comprehensive depiction of cellular processes.

## Figures and Tables

**Figure 1 biomimetics-09-00432-f001:**
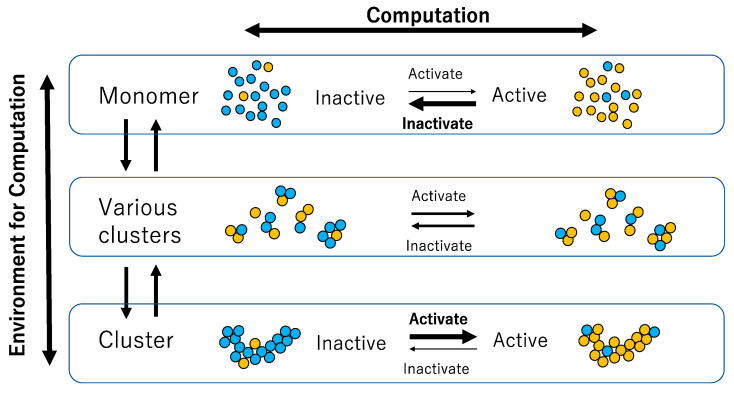
Schematic diagram showing the relationship between chemical reaction (computation) and cluster size (environment for computation). Each circle represents a molecule, with blue and yellow molecules representing inactive and active molecules, respectively. Thick arrows represent dominant reactions under a given environment (i.e., the size of the cluster).

**Figure 2 biomimetics-09-00432-f002:**
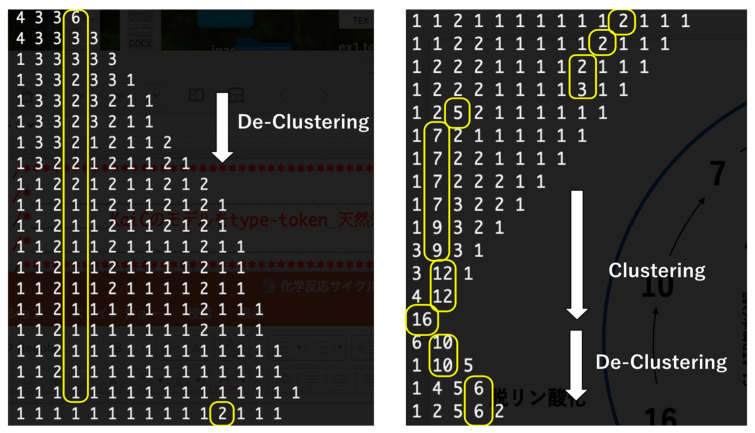
Time series of de-clustering and clustering of molecules. The horizontal line represents the isochron line, and time proceeds from top to bottom and from the left diagram to the right diagram. Each number represents a cluster whose value is the number of molecules contained in a cluster. The yellow loop shows a cluster in which a tracked molecule is contained.

**Figure 3 biomimetics-09-00432-f003:**
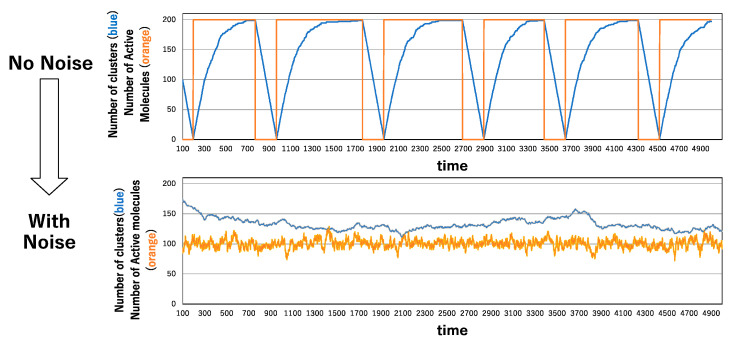
Time series of the number of clusters and the number of active molecules under the no-noise condition (**above**) and under the noise condition of $p_{noise}$ = 0.02 (**bottom**).

**Figure 4 biomimetics-09-00432-f004:**
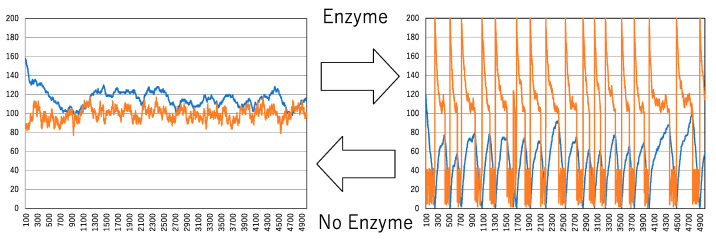
Spike oscillation derived by enzyme accelerating inactivation. The noise condition is set by $p_{noise}$ = 0.02 and $P_{\alpha }=0.7$. The vertical and horizontal axes and two time series colored orange and blue are the same as those in Figure 3.

**Figure 5 biomimetics-09-00432-f005:**
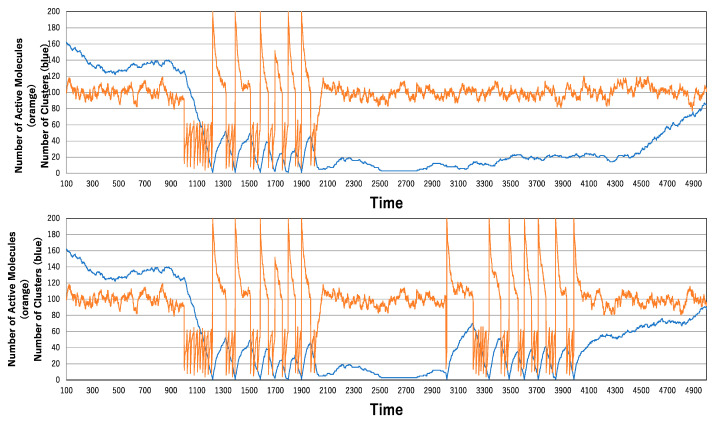
Generation of a spike in the number of active molecules derived by temporal administration of the enzyme: one-shot administration (**above**) and continuous administration over time (**below**).

**Figure 6 biomimetics-09-00432-f006:**
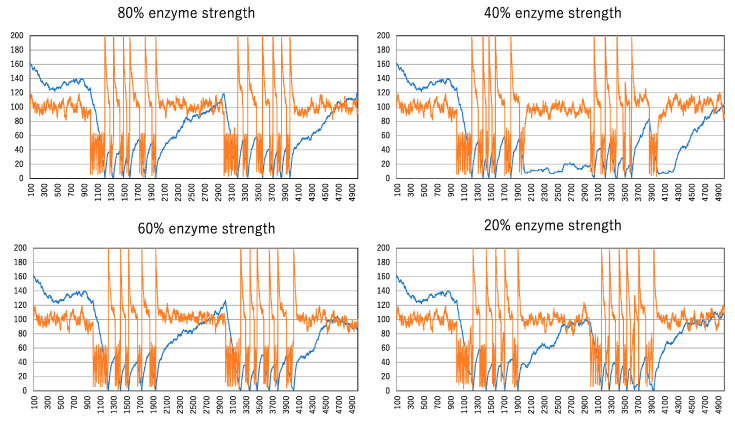
The effect of spike formation is due to the differences in enzymatic strength. The vertical and horizontal lines indicated by orange and blue curves are the same as those in Figure 5.

**Figure 7 biomimetics-09-00432-f007:**
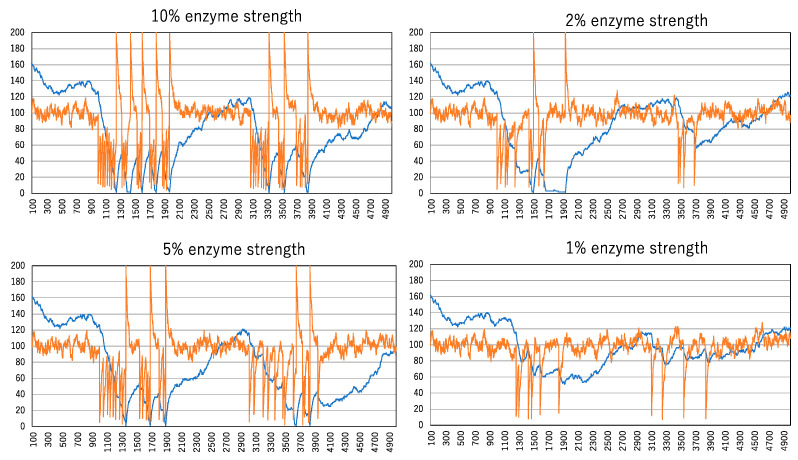
The effect of spike formation is due to the differences in enzymatic strength. The vertical and horizontal lines indicated by orange and blue curves are the same as those in Figure 5.

**Figure 8 biomimetics-09-00432-f008:**
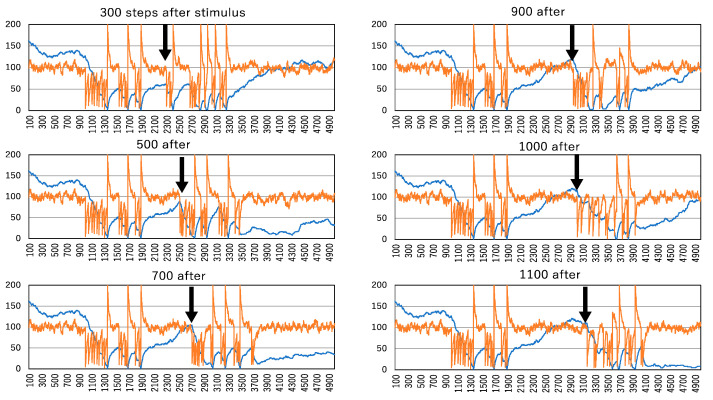
Aftereffect of enzyme administration in the system with an enzyme strength of 2%. At 300, 500, 700, 900, 1000, and 1100 steps after the termination of the first administration of the enzyme, the second enzyme administration is started in 1000 steps. The vertical and horizontal lines indicated by orange and blue curves are the same as those in Figure 5.

**Figure 9 biomimetics-09-00432-f009:**
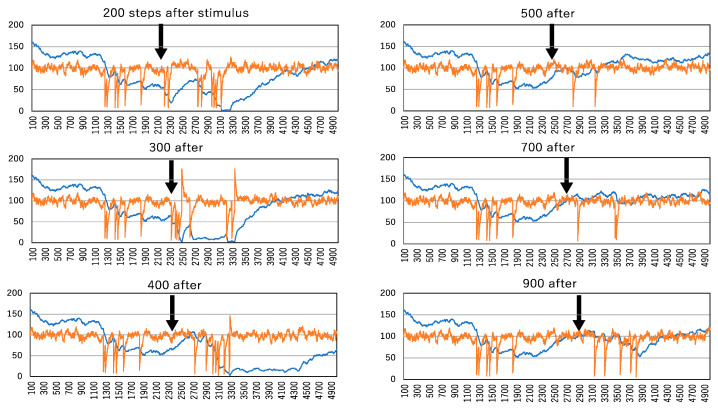
Aftereffect of enzyme administration in the system with an enzyme strength of 1%. At 200, 300, 400, 500, and 900 steps after the termination of the first administration of the enzyme, the second enzyme administration is continued for 1000 steps. The vertical and horizontal lines indicated by orange and blue curves are the same as those in Figure 5.

**Figure 10 biomimetics-09-00432-f010:**
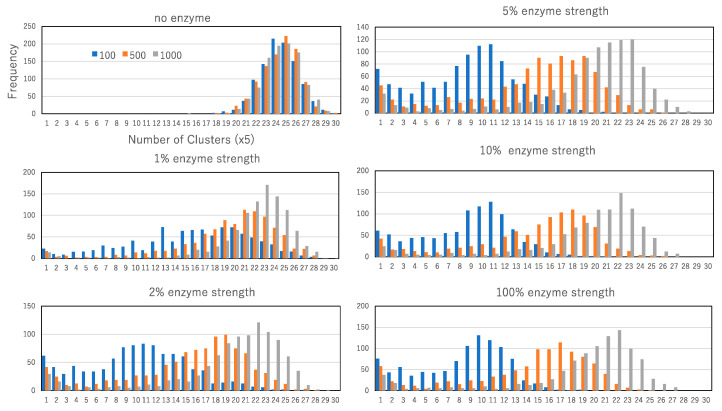
The number of cluster distributions resulting from the aftereffect of enzyme administration with 1, 2, 5, 10, and 100% enzyme strengths compared to the number of cluster distributions resulting from the no-enzyme condition (left top). For each number of cluster distributions from the aftereffect, blue, orange, and gray histograms show the distribution 100, 500, and 1000 steps after the termination of the enzyme administration condition. Histograms are obtained by 1000 trials.

**Figure 11 biomimetics-09-00432-f011:**
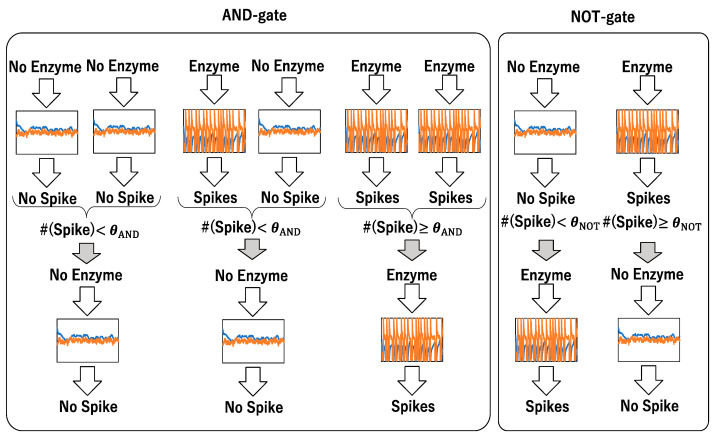
Schematic diagram of AND-gate and NOT-gate implemented by the interaction of chemical reaction (activation) and its environment (clustering of molecules). Each graph represents a time series of the number of clusters (blue) and of the number of active molecules. Under the no-enzyme condition, a fluctuated steady state is obtained, and under the enzyme administration condition, a spike train appears. Counting the number of spikes can implement logic gates.

**Figure 12 biomimetics-09-00432-f012:**
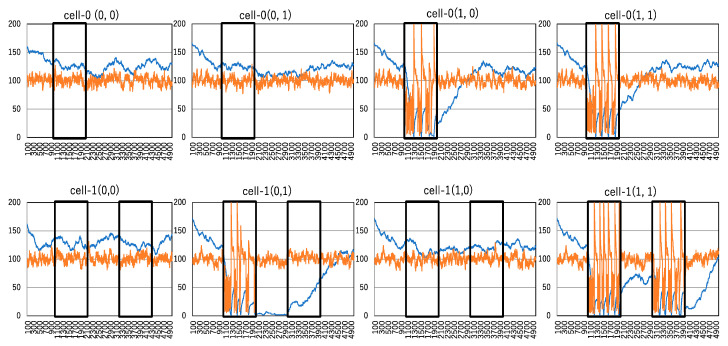
Implementation of AND-gate. A pair of two graphs (cell-0 and cell-1) in the same column represents a pair of inputs, where the input in cell-0 is expressed as a time series of active molecules surrounded by a black square, the input in cell-1 is expressed as a time series of active molecules surrounded by a black square on the left, and the output shown in cell-1 is expressed as a time series of active molecules surrounded by a black square on the right. The vertical and horizontal lines indicated by orange and blue curves are the same as those in Figure 5.

## Data Availability

The datasets presented in this article are not readily available because the data are part of an ongoing study.

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
