# Peer review of "Computation Implemented by the Interaction of Chemical Reaction, Clustering, and De-Clustering of Molecules"

_biomimetics, 2024, doi:10.3390/biomimetics9070432_

Round 1
Reviewer 1 Report
Comments and Suggestions for Authors
Review of the manuscript
“Computation Implemented by Interaction between Chemical Reaction and its Environment in Confined Spaces”
by Yukio Pegio Gunji and Andrew Adamatzky .
Although the title and introduction give the impression that this is an article about the effect of computer radiation on the living system, in fact this is an article about testing a model of cellular automata based on the Gillespie algorithm, to which the authors added the molecule self-aggregation mechanism.
Using model, the authors tried to simulate the effect of external influences on chemical reactions: activation of molecules and the process of clusterization. The authors considered various scenarios of external influence.
It is not clear on what basis the authors decided that this model is suitable for describing the reaction of a living organism to radiation from computers. In the form in which the material is presented, the conclusions of the article are not consistent with the evidence.
It is worth noting that the main part of the article describing the process of testing the model is of interest.
Therefore, in order for it to be accepted for publication, the authors should rewrite the introduction and objectives of the work, putting the model itself and the modification they made at the forefront. Well, you can include one sentence in the discussion that the model can interpret the response of a living organism to some negative influence
Author Response
All replies are contained in the attachment.

Reviewer 2 Report
Comments and Suggestions for Authors
The manuscript "Computation Implemented by Interaction between Chemical Reaction and its Environment in Confined Spaces," by Yukio Pegio Gunji and Andrew Adamatzky,is an excellent reinterpretation of fundamental aspects of chemical reactions and enzyme dynamics. The authors discuss the oscillations and computations that emerge from the interaction between a chemical reaction and its environment, and the effect of enzyme introduction, and explore their potential for contributing to chemical intelligence.
In Section 4, when dealing with the connection to the external environment "as the core of intrinsic intelligence" one is reminded Karl Friston approach to Markov blankets and the minimization of free energy as a central principle of life.
Also, the tradeoff between universality and efficiency reminds Michael Conrad's tradeoff between evolvability, efficiency, and programmability.
Maybe the authors could briefly address these two suggestions in their discussions / conclusions.
I any event, the claim to connect chemical reactions and the effect of introducing enzymes as the crux of intrinsic intelligence versus mechanical intelligence needs more than the customary grain of salt. It is not so easy!
Author Response
All replies to the review are contained in the attachment.

Round 2
Reviewer 1 Report
Comments and Suggestions for Authors
In the paper “Computation Implemented by Interaction between Chemical Reaction and its Environment in Confined Spaces” by Yukio Pegio Gunji and Andrew Adamatzky the authors implement the interaction between a chemical reaction and its environment using an abstract chemical reaction involves only the activation or inactivation of a chemical species with reaction constant changing depending on the environment. The chemical reaction at the molecular level is implemented using the stochastic Gillespie cellular automate algorithm.
The novelty of the research is that the authors make some model experiments involving the influence of noise on the behavior of clustering and de-clustering of molecules and also the enzyme, leading to spike oscillations. The authors have modeled different values of enzyme strengths and evaluated the duration for which the memory of enzyme administration persisted after the initial administration.
It seems to me that the authors take on too much when they assume that the model of cellular automata used, explains the cyclic mechanisms in the life of cells. The article merely explains the operation of the cellular automata model, which does not take into account all possible reactions of a living cell to external influences. But this is just a model; on what basis do the authors believe that the cell reacts to external influences in this way? Why don’t they admit that the memory of an external influence persists for the entire life of the cell? It seems to me that the conclusions of the work should show in what cases the model used can be applied to living cells and to what extent it can be considered reliable. But for this we need at least some comparison with real experiments with appropriate references. At the very least, you need to specify the time in seconds that corresponds to the time step used in the article.
Besides in Figure 5 the lines are almost invisible
The paper needs major revision because of missing controls.
Author Response
The letter to the reviewer is contained in the attachment.

Round 3
Reviewer 1 Report
Comments and Suggestions for Authors
In the paper “Computation Implemented by Interaction between Chemical Reaction and its Environment in Confined Spaces” by Yukio Pegio Gunji and Andrew Adamatzky the authors studied the behavior of a stochastic Gillespie cellular automata model for different values of enzyme strengths and different time intervals of enzyme action.
In the revised version of the article, the authors adjusted the “Discussion” section. A reference to the literature was included that outlined assumptions regarding factors influencing cell distribution. In conclusion, it was noted that the assumption about the course of chemical reactions according to the cellular automata model used, is the authors’ hypothesis.To my mind the conclusions in the text are consistent with the arguments presented and the paper can be published in “Biomimetics” in the present form.